# Rapid Quantitative Detection of Dye Concentration in Pt/TiO_2_ Photocatalytic System Based on RGB Sensing

**DOI:** 10.3390/s25103195

**Published:** 2025-05-19

**Authors:** Cuiyan Han, Ziao Wang, Jiahong Cui, Shuqi Liu, Liu Yang, Yang Fu, Baolin Zhu, Cheng Guo

**Affiliations:** 1School of Pharmacy, Jiangsu Medical College, Yancheng 224005, Chinacuijiahong2004@163.com (J.C.); yangliu240108@163.com (L.Y.); fuyang030514@163.com (Y.F.); 2College of Chemistry, Nankai University, Tianjin 300071, China; 3National Demonstration Center for Experimental Chemistry Education, Nankai University, Tianjin 300071, China

**Keywords:** red–green–blue sensing, Pt/TiO_2_ photocatalyst, methylene blue degradation, partial least squares regression, rapid detection

## Abstract

This article presents an integrated strategy that couples high-efficiency photocatalytic degradation with low-cost, rapid detection to overcome the main drawbacks of conventional TiO_2_-based photocatalysts, including a weak visible-light response, rapid charge–carrier recombination, and reliance on expensive instrumentation for dye concentration detection. Platinum-decorated TiO_2_ (Pt/TiO_2_) was prepared by photoreduction deposition, and systematic characterization confirmed the successful loading of zero-valent Pt nanoparticles onto the TiO_2_ surface, significantly improving charge separation and extending absorption into the visible region. Methylene blue degradation was quantified under ultraviolet (UV) and simulated sunlight; radical-scavenging tests clarified the reaction pathway. In parallel, smartphone images of the reaction mixture were processed in ImageJto extract red–green–blue (RGB) values, which were related to dye concentration through a partial least-squares (PLS) model validated against reference UV–Vis data. Pt/TiO_2_ removed 95.0% of methylene blue within 20 min of UV irradiation and 90.2% within 160 min of simulated sunlight—31.8% and 19.1% faster, respectively, than pristine TiO_2_. The RGB-based PLS model achieved a coefficient of determination (R^2^) of 0.961 for the prediction set. By integrating photocatalysis with smartphone-based colorimetry, the proposed method enables rapid monitoring of organic dyes concentration, providing an intelligent and economical platform for industrial wastewater treatment.

## 1. Introduction

Industrial dye effluents represent a growing environmental and public-health concern, because many organic dyes are toxic to aquatic organisms and humans even at trace levels [1,2]. Consequently, the development of efficient, rapid, and accurate methods for both dye degradation and concentration monitoring have become a key research focus in the field of environmental management.

Photocatalytic degradation technology has garnered widespread attention due to its high efficiency, environmental friendliness, and low energy consumption [1,3]. Among various photocatalysts, titanium dioxide (TiO_2_) is extensively applied in dye wastewater treatment owing to its chemical stability, low toxicity, and cost-effectiveness [4]. However, TiO_2_ absorbs only ultraviolet (UV) light, and the fast recombination of photogenerated electrons and holes significantly limits its photocatalytic efficiency [5]. Noble-metal deposition—particularly with platinum (Pt) [6], gold (Au) [7], or silver (Ag) [8]—has therefore been used to enhance charge separation and extend the optical response into the visible region. Among these options, Pt/TiO_2_ exhibits the greatest performance gain, because metallic Pt effectively captures photogenerated electrons, preventing electron–hole recombination and thereby enhancing catalytic efficiency [6,9]. Moreover, metallic Pt nanoparticles can induce localized surface plasmon resonance (LSPR) effects under visible-light irradiation, which significantly enhances visible-light absorption and consequently promotes photocatalytic activity, thereby further broadening the applicability of photocatalytic technology [6].

Although photocatalysis efficiently removes dyes, it does not directly provide real-time monitoring or quantitative analysis of dye concentrations. Conventional analytical techniques such as UV–Vis spectrophotometry or high-performance liquid chromatography (HPLC) are precise, but require costly, laboratory-based instrumentation that is unsuitable for on-site or high-throughput monitoring. Integrating photocatalytic degradation with a simple and low-cost sensing strategy would therefore enable rapid and accurate monitoring of organic dye concentrations, providing timely and reliable data support for industrial process optimization and emission management.

Smartphone colorimetry—based on red, green, and blue (RGB) channel analysis coupled with a partial least squares (PLS) regression model—has recently emerged as a promising tool for quantitative chemical sensing [10,11,12]. By extracting RGB intensities from a single image, dye concentration can be predicted rapidly without specialized equipment, making the technique attractive for field applications. 

The present study combined Pt/TiO_2_ photocatalysis with an RGB-based PLS detection scheme to achieve rapid methylene blue (MB) concentration monitoring. A Pt/TiO_2_ photocatalyst was synthesized by photoreduction and systematically characterized using X-ray diffraction (XRD), UV–Vis diffuse reflectance spectroscopy (UV–Vis DRS), X-ray photoelectron spectroscopy (XPS), and scanning electron microscopy (SEM). Photocatalytic degradation kinetics were determined under UV and simulated sunlight, while RGB data acquired with a smartphone were correlated with MB concentration through a calibrated PLS model validated against UV–Vis measurements. This innovation—high-efficiency Pt/TiO_2_ photocatalysis combined with instrument-free RGB colorimetry—provides a rapid, low-cost platform for dye-wastewater treatment and rapid quantitative monitoring. 

## 2. Materials and Methods

### 2.1. Reagents

TiO_2_ nanopowder and MB were purchased from Aladdin Biochemical Technology Co., Ltd. (Shanghai, China); chloroplatinic acid, isopropanol, triethanolamine, and anhydrous ethanol were obtained from Guangfu Technology Development Co., Ltd. (Tianjin, China).

### 2.2. Preparation and Characterization of Pt/TiO_2_ Photocatalyst

#### 2.2.1. Preparation of Pt/TiO_2_ Photocatalyst

The Pt/TiO_2_ photocatalyst was synthesized using the photoreduction deposition method [13]. Considering both cost and environmental factors, the Pt loading was deliberately limited to 1.0 wt%. Briefly, 0.99 g of TiO_2_ powder and 0.01 g of chloroplatinic acid were dispersed in 100 mL of an ethanol–water solution (volume ratio 1:1). The resulting suspension was then stirred and irradiated under a 300 W mercury lamp for 4 h at room temperature. During irradiation, photogenerated electrons in the conduction band of TiO_2_ reduced Pt^4+^ ions to Pt^0^ nanoparticles, which subsequently deposited uniformly onto the TiO_2_ surface. After irradiation, the final product was washed thoroughly with deionized water, filtered to remove residual chloride ions (Cl^−^), and dried at 80 °C for 12 h to obtain the Pt/TiO_2_ photocatalyst.

#### 2.2.2. Characterization of Pt/TiO_2_ Photocatalyst

In this study, multiple analytical techniques were employed to comprehensively characterize the Pt/TiO_2_ photocatalyst. The crystal structure was determined using XRD (D/MAX-2500, Rigaku, Tokyo, Japan), while optical response properties were evaluated via UV–Vis DRS (Lambda 750, PerkinElmer, Waltham, MA, USA). The elemental composition and chemical states were analyzed using XPS (Axis Ultra DLD, Kratos Analytical Ltd., Manchester, UK), with the binding energy of adsorbed carbon at 284.6 eV used as a reference for charge correction. The morphology and elemental distribution were examined using SEM (JSM-7800F, JEOL Ltd., Tokyo, Japan).

Photocatalytic degradation experiments were conducted using a photocatalytic reactor (XPA-7 series, Xujiang Electromechanical Plant, Nanjing, China). Images of MB solutions undergoing photocatalytic degradation were captured using a smartphone, and RGB values were extracted and processed using ImageJ software (1.52a).

### 2.3. Photocatalytic Performance and Mechanistic Investigation of Pt/TiO_2_ Photocatalyst

Photocatalytic degradation experiments were conducted using MB as the model pollutant to investigate the photocatalytic performance of Pt/TiO_2_. Irradiation was performed using either a 300 W mercury lamp (UV light) or a 500 W xenon lamp (simulated sunlight). Pure TiO_2_ nanopowder was employed as a reference material to evaluate the enhanced photocatalytic activity of the Pt/TiO_2_ photocatalyst. All photocatalytic experiments were conducted in triplicate to ensure reproducibility.

#### 2.3.1. Photocatalytic Performance Testing

In a typical experiment, 50 mg of either pure TiO_2_ nanopowder or Pt/TiO_2_ photocatalyst was dispersed into 60 mL of a MB solution (54 μM). The resulting suspension was magnetically stirred in the dark for 30 min to establish adsorption–desorption equilibrium. After equilibration, 1.0 mL of the solution was collected, centrifuged, and the absorbance of the supernatant was measured spectrophotometrically to determine the initial concentration (C_0_). Subsequently, the light source (UV or simulated sunlight) was activated, and aliquots were withdrawn at predetermined time intervals. The absorbance of each collected sample was analyzed, and the corresponding MB concentration (C) was determined. The photocatalytic degradation efficiency was evaluated by plotting the relative concentration ratio (C/C_0_) against irradiation time (t).

The recycling performance of the Pt/TiO_2_ photocatalyst was investigated through five consecutive recycling tests under identical conditions (UV irradiation), and the photocatalyst was recovered after each cycle, washed, and reused in five consecutive degradation trials.

#### 2.3.2. Mechanistic Investigation Testing

To identify the primary reactive species involved in the photocatalytic degradation process, radical trapping experiments were performed. Triethanolamine, isopropanol, and nitrogen gas (N_2_) were used as scavengers for photogenerated holes (h^+^), hydroxyl radicals (·OH), and superoxide radicals (·O_2_^−^), respectively. Experimental procedures for radical trapping were identical to those described in Section 2.3.1, with the respective scavengers introduced before initiating irradiation.

### 2.4. RGB-Based PLS Model for Quantitative Detection of Methylene Blue Concentration

Initially, a series of MB standard solutions with concentrations ranging from 0 to 54 μM (with 6 μM intervals) were prepared. Each standard solution was transferred onto a white ceramic well plate. To minimize variations due to lighting and camera positioning, stringent image acquisition protocols were adopted: the smartphone camera was fixed at a consistent distance and angle, ambient lighting was kept uniform, and a neutral background was used for all photographs. 

Subsequently, RGB channel values were extracted from the captured images using ImageJ software. Extracted RGB data and corresponding MB concentrations were imported into the PLS toolbox in MATLAB (R2024a) to establish a quantitative calibration model correlating RGB values with MB concentrations. The predictive performance of the calibration model was assessed by cross-validation, determining key parameters such as the root mean square error of calibration (RMSEC), root mean square error of cross-validation (RMSECV), and correlation coefficient (R^2^). 

Finally, images of MB solutions undergoing photocatalytic degradation were taken under identical imaging conditions, and RGB values were extracted. These RGB data were then input into the previously established PLS calibration model to predict MB concentrations. The concentrations predicted using the RGB-based PLS model were validated by comparing them against results obtained with a standard UV–Vis spectrophotometer.

## 3. Results and Discussion

### 3.1. SEM Analysis

Figure 1 displays SEM micrographs of the Pt/TiO_2_ photocatalyst. Aggregated nanoparticles were observed, and the individual Pt particles (approximately 3–5 nm) were too small to be resolved directly. Nevertheless, the successful loading of Pt nanoparticles onto the TiO_2_ nanopowder was confirmed through elemental mapping and UV–Vis DRS, and XPS data.

### 3.2. Structural and Optical Characterization (XRD, UV–Vis DRS, and XPS)

The crystal structure of the Pt/TiO_2_ photocatalyst was analyzed using XRD. The XRD image (Figure 2A) shows only reflections of anatase TiO_2_ (JCPDS No. 21-1272), with diffraction peaks at 25.28°, 37.82°, 48.12°, 53.94°, 55.18°, 62.82°, 68.86°, 70.36°, and 75.26°, corresponding to the (101), (004), (200), (105), (211), (204), (116), (220), and (215) crystal planes, respectively. These peaks align well with the standard reference card, indicating the high crystallinity of the synthesized photocatalyst. No additional peaks were detected, indicating that Pt deposition did not alter the host crystal phase and that the photocatalyst was phase-pure and highly crystalline.

Furthermore, XPS analysis was conducted to investigate the chemical state of Pt in the Pt/TiO_2_ photocatalyst. As shown in Figure 2C, wide-scan XPS spectra confirm the successful incorporation of Pt on the TiO_2_ surface, and no discernible shifts are observed in the Ti 2p region, suggesting that Ti^4+^ cations remained stable during photoreduction [14]. High-resolution Pt 4f spectra (Figure 2D) reveal Pt 4f_7/2_ (~70.5 eV) and Pt 4f_5/2_ (~73.8 eV) peaks characteristic of metallic Pt^0^. The narrow width of these peaks, combined with their alignment with reported reference values for metallic platinum, further verify the complete photoreduction of Pt^4+^ to Pt^0^ and the uniform dispersion on the support [15,16]. UV–Vis DRS (Figure 2B) further reveals that, compared to pure TiO_2_ (absorption edge at 400 nm, bandgap of 3.2 eV), Pt/TiO_2_ exhibited a significant redshift of the absorption edge and broad-spectrum absorption enhancement in the 500–800 nm visible light region. This broadening is attributed to the LSPR effect of Pt^0^ nanoparticles. Additionally, the XPS-confirmed metallic Pt^0^ chemical state provides direct electronic structural evidence supporting the LSPR-induced visible-light response mechanism.

Collectively, these characterization results indicate that Pt^0^ was homogeneously distributed on anatase TiO_2_, simultaneously improving charge–carrier separation and extending optical absorption via the LSPR effect. These features provide a solid basis for high-efficiency visible-light-driven photocatalysis.

### 3.3. Photocatalytic Performance and Mechanistic Investigation of Pt/TiO_2_

The photocatalytic performance of the Pt/TiO_2_ photocatalyst was evaluated using an MB solution as a target pollutant. Figure 3A shows the degradation profiles of the MB solution under UV irradiation in the presence of Pt/TiO_2_ and TiO_2_. The inset photograph clearly illustrates the progressive color change of the MB solution from dark blue to nearly colorless, visually confirming efficient dye removal. Specifically, Pt/TiO_2_ achieved a degradation efficiency of 95.0% within 20 min, markedly superior to the 63.2% observed for pure TiO_2_. Figure 3B shows the corresponding first-order kinetic plots (ln(*C*_0_/*C*) vs. time), confirming that MB degradation follows a first-order kinetic [17] as described by the Langmuir–Hinshelwood model:$$\ln\frac{C_{0}}{C}=kt$$
where *k* represents the degradation rate constant. The calculated rate constants for Pt/TiO_2_ and TiO_2_ were 0.0655 min^−1^ and 0.0584 min^−1^, respectively, demonstrating that Pt/TiO_2_ significantly enhances photogenerated charge–carrier separation efficiency compared to unmodified TiO_2_.

Radical trapping experiments were conducted to identify the primary reactive species involved in MB degradation, and results are presented in Figure 3C. The addition of triethanolamine (a scavenger of photogenerated holes, h^+^) significantly suppressed degradation efficiency, indicating that h^+^ was the predominant reactive species. Introducing nitrogen gas (an ·O_2_^−^ scavenger) partially inhibited degradation, suggesting superoxide radicals (·O_2_^−^) served an auxiliary role. Conversely, adding isopropanol (a hydroxyl radical, ·OH, scavenger) had minimal effect, implying a minor role for ·OH radicals.

Based on these observations, a plausible photocatalytic mechanism is proposed in Figure 3D. Under UV irradiation, electrons from the valence band (VB) of TiO_2_ migrate to the conduction band (CB), generating electron–hole pairs. Photogenerated electrons subsequently react with adsorbed molecular oxygen to form reactive oxygen species, such as superoxide radicals (·O_2_^−^) and hydrogen peroxide (H_2_O_2_), facilitating MB degradation. Due to the high Schottky barrier at the Pt–TiO_2_ interface, electrons transfer efficiently to Pt nanoparticles, significantly suppressing electron–hole recombination and enhancing overall catalytic activity [18]. Simultaneously, photogenerated holes (h^+^) directly oxidize MB molecules, ultimately decomposing them into non-toxic products such as CO_2_ and H_2_O.

Reusability is crucial for practical wastewater treatment applications. Thus, the recycling performance of the Pt/TiO_2_ photocatalyst was examined over five consecutive degradation cycles under identical UV irradiation (Figure 4A,B). After each cycle, the catalyst was recovered, washed, and reused. Degradation efficiencies remained above 85% even after the fifth reuse, indicating excellent reusability of the Pt/TiO_2_ photocatalyst.

Photocatalytic performance was further assessed under simulated sunlight conditions (Figure 5A). Pt/TiO_2_ achieved a degradation efficiency of 90.2% after 160 min, significantly higher than the 71.1% observed for pure TiO_2_. Additionally, kinetic analysis confirmed that the rate constant for Pt/TiO_2_ was approximately 1.9 times higher than that of TiO_2_, with excellent fitting quality (R^2^ = 0.9703 for Pt/TiO_2_ and R^2^ = 0.9775 for TiO_2_; Figure 5B). The enhanced photocatalytic efficiency under visible-light conditions is attributed primarily to the LSPR effect of metallic Pt nanoparticles, facilitating improved absorption and utilization of visible-light energy [19,20]. While pure TiO_2_ was the primary control in this study, Pt was specifically selected due to its established capacity to enhance electron–hole separation and broaden visible-light absorption through the LSPR effect. Although Au- and Ag-modified TiO_2_ also exhibit enhanced photocatalytic performance according to previous reports, the choice of Pt was guided by its superior catalytic stability and extensively documented efficiency in similar applications [21,22].

### 3.4. Rapid Detection of Methylene Blue Concentration Using RGB-Based Sensing Model

Figure 6A presents the absorbance–concentration calibration curve obtained for MB solutions. Within the investigated concentration range, the absorbance of MB demonstrates an excellent linear correlation with its concentration. To enable rapid quantitative detection, an RGB-based PLS model was established using RGB channel values extracted from smartphone images of standard MB solutions. The optimal number of latent variables in the PLS model was determined by performing leave-one-out cross-validation and selecting the number corresponding to the minimum root mean square error of cross-validation (RMSECV).

The final PLS calibration model for predicting MB concentrations utilized three latent variables, yielding a root mean square error of calibration (RMSEC) of 2.513 and a correlation coefficient (R_c_^2^) of 0.979. Cross-validation results provided an RMSECV of 3.860 and a correlation coefficient (R_cv_^2^) of 0.951. Additionally, the residual predictive deviation (RPD) value of the model was 4.7 (RPD > 3), indicating excellent predictive capability.

To further verify the model’s predictive accuracy, an independent prediction dataset was employed and the model’s accuracy was evaluated using the root mean square error of prediction (RMSEP) and the prediction correlation coefficient (R_p_^2^). As shown in Figure 6B, the model produced a root mean square error of prediction (RMSEP) of 3.736 and a correlation coefficient (R_p_^2^) of 0.961, demonstrating robust reliability and strong linear correlation predictive performance.

Multiple measurements performed under uniform lighting conditions on MB solutions of identical concentrations produced minimal variation in RGB values, confirming excellent reproducibility of the image acquisition protocol and analytical procedure. As this sensing approach involves non-destructive imaging with only a smartphone camera and standard image-processing software (e.g., ImageJ), the RGB-based method can be reused indefinitely without substantial additional costs. Additionally, repeated analyses can be readily conducted by simply replacing the solution in a standard well plate, demonstrating practical suitability for continuous or repeated monitoring.

Compared to traditional UV–Vis spectrophotometric methods, the RGB-based PLS model significantly simplifies sample handling and reduces both material and labor costs. This methodology offers a convenient, low-cost, and instrument-free alternative for the rapid quantitative detection of organic dye concentrations. If further developed into a dedicated mobile application, this approach would allow users to capture a single smartphone image of dye-containing wastewater, enabling immediate concentration determination without complex analytical procedures. Such a capability holds strong practical promises for routine monitoring and management of organic dyes in industrial wastewater.

## 4. Conclusions

This study successfully demonstrated a rapid detection method for detecting MB concentrations using a smartphone-based RGB sensing approach combined with a PLS model. The developed sensing method was effectively applied to monitor the degradation of MB using a Pt/TiO_2_ photocatalyst. Characterization results confirmed the uniform dispersion of zero-valent Pt nanoparticles on the anatase TiO_2_ surface, significantly enhancing the separation efficiency of photogenerated charge carriers and expanding the photocatalyst’s absorption into the visible region through LSPR. The proposed RGB-based PLS sensing approach eliminates reliance on costly analytical instruments, providing an efficient, low-cost, and easily implementable solution for rapid on-site monitoring of organic dyes in industrial wastewater. Furthermore, integrating this method into a dedicated mobile application in the future could substantially advance the synergistic combination of photocatalytic wastewater treatment and intelligent detection technologies. 

## Figures and Tables

**Figure 1 sensors-25-03195-f001:**
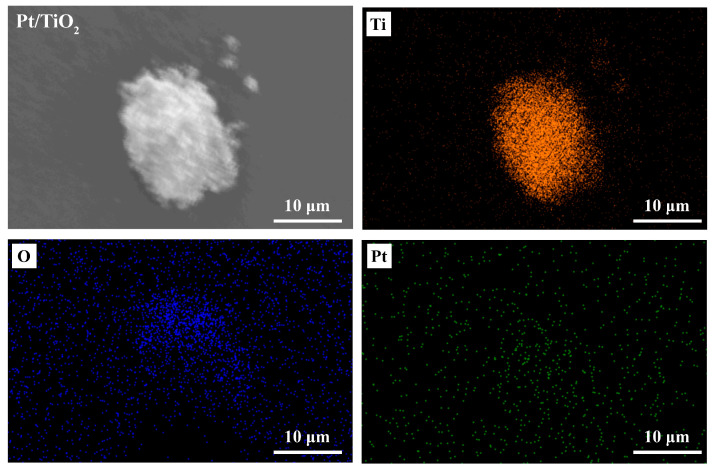
SEM images and elemental mapping of Pt/TiO_2_ photocatalyst.

**Figure 2 sensors-25-03195-f002:**
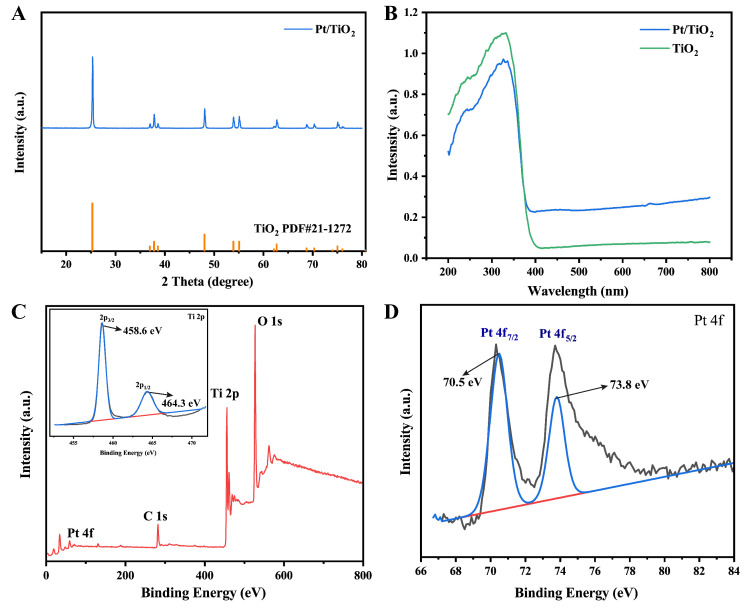
(**A**) XRD spectra of Pt/TiO_2_ photocatalyst; (**B**) UV–Vis DRS spectra of TiO_2_ and Pt/TiO_2_ photocatalysts; (**C**) XPS wide-spectrum of Pt/TiO_2_ photocatalyst and corresponding narrow scan spectra of Ti 2p; and (**D**) corresponding XPS narrow scan spectra of Pt 4f.

**Figure 3 sensors-25-03195-f003:**
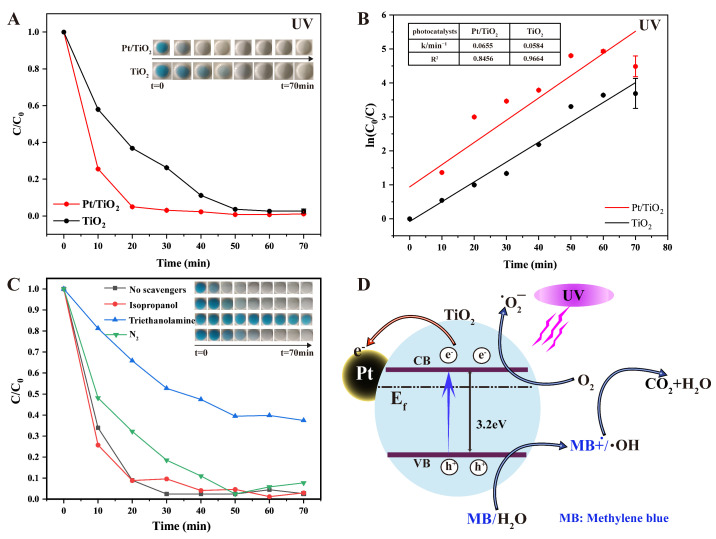
(**A**) Photocatalytic degradation of MB by TiO_2_ and Pt/TiO_2_ under UV irradiation (inset images present photocatalytic degradation of MB solution at different times); (**B**) first-order kinetics of MB degradation under UV irradiation in TiO_2_ and Pt/TiO_2_ systems at different times (inset tables present rate constants and R^2^ values of all photocatalysts); (**C**) photocatalytic degradation of MB upon addition of various scavengers under UV irradiation in Pt/TiO_2_ system; and (**D**) proposed photocatalytic mechanism of Pt/TiO_2_ system under UV irradiation.

**Figure 4 sensors-25-03195-f004:**
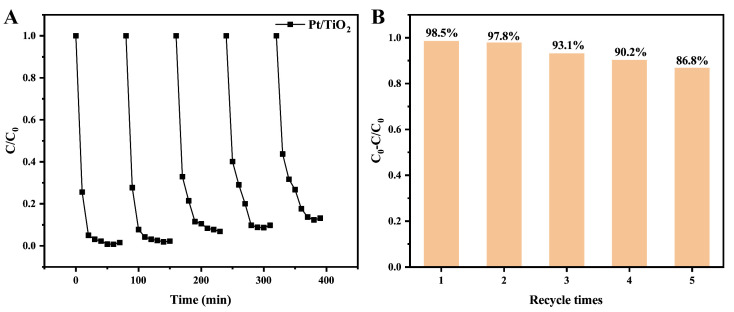
Recyclability of Pt/TiO_2_ toward the degradation of MB under UV irradiation. (**A**) the variation of the ratio C/C_0_ with time during the UV-induced degradation of MB over Pt/TiO_2_; (**B**) the degradation efficiency of Pt/TiO_2_ for MB under UV irradiation after different recycle times.

**Figure 5 sensors-25-03195-f005:**
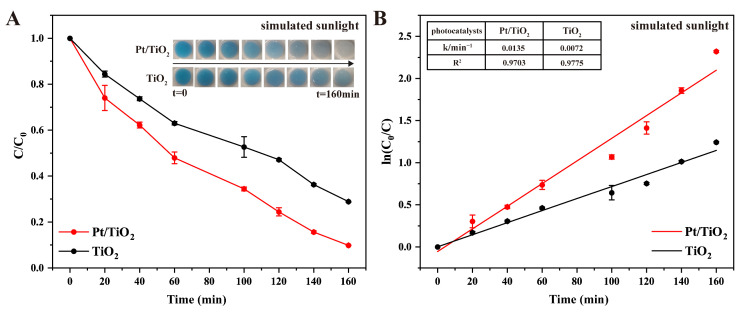
(**A**) Photocatalytic degradation of MB by TiO_2_ and Pt/TiO_2_ under simulated sunlight irradiation (inset images present photocatalytic degradation of MB at different times), and (**B**) first-order kinetics of MB degradation under simulated sunlight irradiation in TiO_2_ and Pt/TiO_2_ systems at different times (inset table presents rate constants and R^2^ values of all photocatalysts).

**Figure 6 sensors-25-03195-f006:**
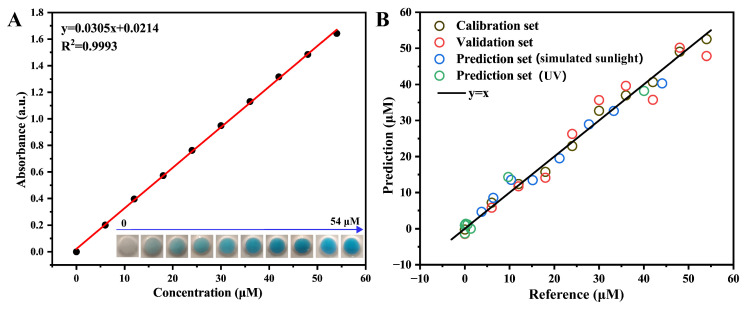
(**A**) Absorbance–concentration standard curve of MB (inset photos present standard solutions of MB with concentrations ranging from 0 to 54 μM); and (**B**) RGB-based PLS model for rapid monitoring of MB concentrations.

## Data Availability

Data presented in this study are openly available upon request.

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
