# Peer review of "Rapid Quantitative Detection of Dye Concentration in Pt/TiO2 Photocatalytic System Based on RGB Sensing"

_sensors, 2025, doi:10.3390/s25103195_

Round 1

Reviewer 1 Report

Comments and Suggestions for Authors

The research is meaningful, and falls within the scope of this journal. I suggest the following revisions:

  1. The error bars should be given in Figure 4. Also, the R2 values are also needed.
  2. What is the meaning of RGB? A more mention may be needed.
  3. The stability and reusability of the sensing.

Reviewer 2 Report

Comments and Suggestions for Authors

In this study, the combination of Pt/TiO2 photocatalytic degradation and rapid detection based on RGB-PLS modeling effectively solved the problems of weak visible light response, high carrier complexation rate, and dependence on expensive instrumentation for dye concentration detection of traditional TiO2 photocatalysts. The logic of the article is clear and the experimental design is more reasonable, which has certain potential application value. However, some of the experimental details and validation in practical application scenarios still need further improvement. Therefore, I recommend that the study be reconsidered with major revisions. The detailed comments are as follows:

  1. Only pure TiO2 was used as a control, and the performance of modified TiO2 was not compared with that of other precious metals (e.g., Au, Ag), which could not fully reflect the superiority of Pt.
  2. Pt, as a precious metal, is relatively expensive, and its economy and environmental friendliness for large-scale application need to be evaluated.
  3. The recycling performance or long-term stability of Pt/TiO2 has not been investigated, please supplement the related experiments.
  4. In the photocatalytic degradation experiments, were the effects of different concentrations, different initial pH values and other factors on the degradation effect considered? These factors may significantly affect the rate and efficiency of the photocatalytic reaction and need to be further explored.
  5. The experiments were conducted only for the pure solution of a single dye (MB), and the interference of coexisting pollutants (e.g., organics) in the actual wastewater on the photocatalysis and detection was not tested.
  6. Were the environmental conditions (e.g., light intensity, angle, etc.) of image acquisition strictly controlled during the RGB-PLS modeling process? Are the differences between different smartphone cameras taken into account? The image sensors and imaging algorithms of different devices may lead to differences in RGB values.

Reviewer 3 Report

Comments and Suggestions for Authors

Smartphone camera and PLS model for monitoring degradation of methylene blue using Pt/TiO2 catalysts were used. The RGB-PLS data were validated by spectrophotometric measurements.

Please state de novelty of the manuscript!

Comparison with other optical sensors should be included in the manuscript.

Page 5: Please explain: “…high-resolution XPS spectra of the Pt 4f orbital …. confirm that Pt exists in the metallic state (Pt⁰) and is well dispersed on the TiO₂ surface”.

No XPS fitting (Pt 4f) was supplied. Are the Ti4+ cations partially reduced?

Errors for calculation of the degradation rate were not included in the manuscript. Figures 3A and C have not the same x-axis.

Round 2

Reviewer 2 Report

Comments and Suggestions for Authors

After revisions, the completeness and self-sufficiency of some details in the revised MS have indeed been improved, I recommended that it can be published after adding a reference (Chem. Eur. J. 2024, 30, e202402102) in ref.21 to support Pt role.

Reviewer 3 Report

Comments and Suggestions for Authors

Extensive modifications were performed.

The first sentence of the Introduction is duplicated.

No plasmon aspect of Pt was discussed in the introduction.

Instrumental characterization should be performed after the material has been prepared.

Initial concentration C0 was spectroscopically obtained.

Was the Pt content of the samples checked?
